# Optimal Strategies for Federated Learning Maintaining Client Privacy

## Abstract

Federated Learning (FL) emerged as a learning method to enable the server to train models over data distributed among various clients. These clients are protective about their data being leaked to the server, any other client, or an external adversary, and hence, locally train the model and share it with the server rather than sharing the data. The introduction of sophisticated inferencing attacks enabled the leakage of information about data through access to model parameters. To tackle this challenge, privacy-preserving federated learning aims to achieve differential privacy through learning algorithms like DP-SGD. However, such methods involve adding noise to the model, data, or gradients, reducing the model's performance.

This work provides a theoretical analysis of the tradeoff between model performance and communication complexity of the FL system. We formally prove that training for one local epoch per global round of training gives optimal performance while preserving the same privacy budget. We also investigate the change of utility (tied to privacy) of FL models with a change in the number of clients and observe that when clients are training using DP-SGD and argue that for the same privacy budget, the utility improved with increased clients. We validate our findings through experiments on real-world datasets. The results from this paper aim to improve the performance of privacy-preserving federated learning systems.

## 1 Introduction

*Federated Learning* (FL) McMahan et al. (2017a) is a distributed method of training ML models where a server shares an initial model among clients who train the model on their locally stored dataset. These locally trained models are then shared with the server, and the process is repeated. FL has been a widely adopted technique for training models for two reasons: (1) Clients want to preserve the privacy of their data from other clients and the server, and (2) Distributing training among many smaller clients is faster and more cost-effective in some cases.

Clients like hospitals and banks possess data that potentially can have a huge impact if ML models are trained using their collective data. Still, they are highly sensitive and cannot be leaked to an adversary. Consider a scenario where multiple hospitals collaborate to train a model for predicting a certain disease using the data they possess. Each hospital can choose to train its models, but the model performance will improve if it is trained collectively. However, certain privacy regulations and risks prohibit hospitals from sharing the data with other branches.

Federated learning (FL) is a solution for collaborative learning in such settings. In FL, all entities, typically called *clients*, share only the trained model (which may be trained using multiple rounds termed *local epochs*) with a *server* who aggregates these models. One iteration of client-server-client model passing is termed as *global rounds*. These steps are iterated for multiple rounds.

Recent advances demonstrated that even model sharing could leak certain information about the data through sophisticated inference attacks Fredrikson et al. (2015); Shokri et al. (2017). Differential Privacy

Dwork (2006) is a technique used to obtain private statistics of a dataset. DP is achieved by adding appropriate noise to statistics on the data which prevents the adversary from inferring the true distribution of the data by introducing randomness. While the noise addition affects the performance of the ML model, it will be difficult to infer the data from these models.

Our goal in this study is to formally study privacy guarantees of `DP-SGD` in FL settings, which is easy by appropriately combining guarantees of `DP-SGD` for single client and composition theorem from Smith et al. (2021). Another important analysis we make is how frequently clients should communicate with the server for the same privacy budget to improve performance. Note that noise addition improves (or maintains) the privacy budget of a given mechanism. If $\mathcal{M}$ is $(\epsilon, \delta)$ private, then $\mathcal{F} \circ \mathcal{M}$ also inherits same (or better) privacy guarantees when $\mathcal{F}$ is independent of the input data. This idea brings a clever realization that Federated averaging is a noise addition step, improving the model performance. Our work leverages this observation to improve the performance of FL models while maintaining privacy guarantees. In short, the best performance is achieved for the same number of total training epochs if we do `FedAvg` after every local training round (Theorem 2). Here best performance means expected loss in $\ell_1$ norm of model parameters w.r.t. without any privacy guarantes. Next, we formally prove that if the total number of clients increases, the FL can get performance closer to that of a model trained without privacy concerns. Thus, for many client settings, we need not worry about the model quality due to noisy training. Lastly, we analyze the effect of local training rounds before global communication and the effect of the number of clients on performance parameter accuracy for different real-world datasets with different architectures with numerous privacy budget combinations. In summary, the following are our contributions.

**Our Contributions.**

We adopt `DP-SGD` Abadi et al. (2016) for individual client training and `FedAvg` McMahan et al. (2017a) for aggregating client model parameters within the FL framework. Training Differentially Private Models for individual clients ensures Local DP. Throughout this paper, we call this framework **PFL**. We show that:

- Given a fixed privacy budget, performance is proportional to the frequency of aggregation step. That means, the clients should update their local model to the server every local epoch for optimal performance. (Theorem 2).

- As the number of participating clients increases, the aggregate global model converges to that of its non-private counterpart (Theorem 3).

- We empirically validate the proofs for Theorem 2 and Theorem 3 by conducting extensive experiments on MNIST, FashionMNIST, CIFAR10 datasets using different DP-SGD techniques Abadi et al. (2016); Papernot et al. (2021); Tramer & Boneh (2021).

## 2 Related Work

Here, we briefly discuss the recent development around Privacy Preserving Federated Learning and highlight the gap in the literature which our work addresses.

**Differential Privacy in Federated Learning** Several recent studies have integrated Differential Privacy (DP) techniques into Federated Learning (FL) frameworks to enhance privacy on the collaborative model trained. Existing works Geyer et al. (2017); McMahan et al. (2017b) ensure client-level privacy against external adversaries using `DP-SGD` in FL. Yu et al. (2020) demonstrated the impact on the performance of the end model for private federated learning and proposed techniques that incentivize local clients to participate in the training process. Balle et al. (2020) proposed a random check-in distributed protocol without the requirement for the data to be uniformly sampled which is not always possible while training in a distributed setting. There have also been other works on privacy-preserving ML without using DP-SGD - Wei et al. (2020a) adds noise to model weights while communicating to the client, Kairouz et al. (2021) adds noise to the gradients in Follow The Regularized Leader Algorithm.

Table 1: Comparison of Related Work

| Reference | Privacy | Privacy Analysis | Utility Analysis | Adversary |
|---|---|:---:|:---:|---|
| Abadi et al. (2016) | Example | ✓ | ✗ | External |
| Geyer et al. (2017) | Client | ✓ | ✗ | External |
| McMahan et al. (2017b) | Client | ✓ | ✗ | External |
| Yu et al. (2020) | Client | ✗ | ✓ | External |
| Kairouz et al. (2021) | Client | ✓ | ✗ | External |
| Wei et al. (2020a) | Client | ✓ | ✗ | External |
| Naseri et al. (2020) | Example | ✗ | ✗ | Server |
| Truex et al. (2020) | Example | ✓ | ✗ | Server |
| **PFL** (this paper) | Example | ✓ | ✓ | Server |

Adversary: $External \subseteq Server$

**Local Differential Privacy in Federated Learning** The mentioned protocols assume a trusted central server and focus on defending against an external adversary. Training Local Differentially Private models and communicating them to the server Naseri et al. (2020) ensures privacy against both the server and external adversaries. Our work focuses on protocols for training optimal models in this setting. We also present theoretical and empirical analysis of our protocols. We compare our work with existing Private Federated Learning methods in Table 1

## 3   Preliminaries

This section presents the background essential to our study of private federated learning. Consider the classification problem as follows. Let $\mathcal{X} \subseteq \mathbb{R}^d$ be the instance space and $\mathcal{Y} = \{1, \ldots, m\}$ be the label space where $m$ is the number of classes. Let $\mathcal{D}$ be the joint distribution over $\mathcal{X} \times \mathcal{Y}$. Training data contains pairs $(\mathbf{x}_i, y_i)$, $i = 1 \ldots n$, where every $(\mathbf{x}_i, y_i) \in \mathcal{X} \times \mathcal{Y}$ is drawn i.i.d. from the distribution $\mathcal{D}$. The objective for the ML algorithm is to learn a vector values function $\mathbf{f} : \mathcal{X} \to \mathbb{R}^m$ using the training data which assigns a score value for each class. For an example $(\mathbf{x}, y)$, the predicted class label is found as $\hat{y} = \arg\max_{j \in [m]} f_j(\mathbf{x})$. The objective here is that $\hat{y}$ should be the same as the actual class label $y$. To ensure that, we use a loss function $\mathcal{L} : \mathbb{R}^m \times [m] \to \mathbb{R}_+$[1], which assigns a nonnegative score depending on how well the classifier predicts the class label for example. We use cross-entropy loss in this work. Learning is done by minimizing cumulative loss achieved by the model $\mathbf{f}$ on the training data.

### 3.1   Federated Learning

In a federated learning setting, $\{c_1, c_2, \ldots, c_k\}$ different entities are engaging in learning the same $f$ from their own data. In this paper, we assume all of them are training a neural network (NN) with the data available locally. If they all cooperate, they can learn $f$ that offers better performance. That is where *federated learning* (FL) plays a crucial role. In FL, there is a server $S$ that coordinates the learning through multiple rounds of communication. It initializes the model $theta^0$[2] and shares with the entities $\{c_1, c_2, \ldots, c_k\}$, commonly referred as *clients*. In general, at the beginning round $r$ it shares a global model $M^{r-1}$ with the clients. Each $c_i$, updates the model $M_i^r$ with its data $D_i$ while training happens for $E_i^r$ *epochs*. We denote the corresponding NN parameters as $\theta_i^r$. All the clients share their model parameters with the server and the server aggregates them as $\theta^r = F_{agg}(\theta_1^r, \ldots, \theta_k^r)$. These rounds continue till some stopping criteria are met. Such a setting is called *homogeneous*, and all clients train the same model (e.g., a neural network with the same architecture, same activation functions and same loss function etc.). In the non-homogenous setting, different clients can use different classifier models.

---

[1]$\mathbb{R}_+$ denotes the set of nonnegative real numbers.
[2]does not matter if it is random or based on some prior knowledge.

In this paper, we consider only the homogenous setting. We assume the global training happens in $R$ global rounds. Thus, the total training epochs (the number of access to $D_i$) is $T = \sum_{r=1}^{R} E_r$. We use $F_{agg} =$`FedAvg` McMahan et al. (2017a) as the aggregation technique. In `FedAvg`, each $c_i$ shares locally trained parameters $\boldsymbol{\theta}_i^r$ with $S$ which computes the central model's parameter as the average of $\boldsymbol{\theta}_i^r$. Thus, $\boldsymbol{\theta}^r = \frac{1}{k} \sum_{i=1}^{k} \boldsymbol{\theta}_i^r$.

In FL, though the clients are not sharing the data, due to model inversion attacks Fredrikson et al. (2015), clients with highly sensitive data, such as hospitals, and banks, may prefer not to participate in collaborative learning. Differential Privacy plays an important role in mitigating privacy concerns. In this work, we consider the clients that train differentially private (DP) models. The next subsection explains DP briefly.

## 3.2 Differential Privacy

*Differential Privacy* (DP) ensures consistent output probabilities for computations, regardless of individual data inclusion or removal.

**Example level DP.** Our work focuses on example-level privacy Truex et al. (2020) in FL systems, compared to client-level privacy Geyer et al. (2017); McMahan et al. (2017b); Kairouz et al. (2021); Wei et al. (2020b). Our objective is to prevent leakage of any individual's data from a single client's dataset, making example-level privacy a natural choice for defining Differential Privacy (DP), as formally described in Definition 1.

**Definition 1 (Differential Privacy Dwork (2006); Dwork et al. (2014))** *A randomized algorithm $\mathcal{A}$ with input $D \subset \mathcal{X}$ is $(\epsilon, \delta)$-differentially private if $\forall O \subseteq Range(\mathcal{A})$ and for all $D, D' \subset \mathcal{X}$ such that $|(D \setminus D') \cup (D' \setminus D)| \leq 1$:*

$$\mathbb{P}[\mathcal{A}(D) \in O] \leq e^\epsilon \cdot \mathbb{P}[\mathcal{A}(D') \in O] + \delta.$$

The parameters $\epsilon$ and $\delta$, collectively known as the *privacy budget*, control the level of privacy protection. Typically, adding zero-mean noise to the query's answer aids in achieving DP. When multiple privacy-preserving mechanisms are working simultaneously, we use *parallel composition* to explain how privacy guarantees accumulate. Below Theorem states the privacy guarantees using parallel composition more formally Smith et al. (2021).

**Theorem 1 (Heterogeneous Parallel Composition (Theorem 2 in Smith et al. (2021)))** *Let $D \subset \mathcal{X}$ be a dataset, and $k \in \mathbb{N}$. , let $D_i \subset D \; \forall i \in [k]$, let $\mathcal{A}_i$ be a mechanism that takes $D \cap D_i$ as input, and suppose $\mathcal{A}_i$ is $\epsilon_i$-DP. If $D_i \cap D_j = \emptyset$ whenever $i \neq j$, then the composition of the sequence $\mathcal{A}_1, \ldots, \mathcal{A}_k$ is $\max\{\epsilon_1, \ldots, \epsilon_k\}$-DP.*

The Parallel Composition Theorem demonstrates the additive nature of privacy parameters $\epsilon$ and $\delta$ in parallel execution scenarios. We next do a formal privacy and utility analysis of `DP-SGD` in FL settings in the following section.

## 3.3 Differentially Private Stochastic Gradient Descent (DP-SGD)

`DP-SGD` Abadi et al. (2016) safeguards the privacy of gradients generated during the stochastic gradient descent optimization process. During the learning process, the gradient of the loss with respect to the model parameters is computed for each example and clipped to a maximum $l_2$ norm of $C$. Then noise sampled from zero-mean multivariate normal distribution with covariance matrix $\sigma^2 C^2 \mathbf{I}$ is added to the average of the gradients corresponding to samples in a mini-batch. This guarantees $(\epsilon, \delta)-$DP on the data by limiting the sensitivity of the output to each data point. This achieved optimal privacy-utility guarantees compared to methods of that time and enabled deploying real-world differentially private systems Apple (2017); Google (2023).

**DP-SGD with Tempered Sigmoid Activation:** Clipping the gradients leads to information loss, especially when we clip large gradients. Papernot et al. (2021) showed that replacing unbounded activation

functions like ReLU in the neural network with a family of bounded activation functions (e.g., tempered sigmoid activation etc.) controls the magnitude of gradients. Tempered sigmoid activation function $\phi_{s,temp,o}(.)$ has the following form.

$$\phi_{s,temp,o}(z) = \frac{s}{1 + e^{-temp \cdot z}} - o$$

Where $s$ represents the scale, $temp$ is the inverse temperature and $o$ is the offset. Tuning these parameters and clipping parameter $C$ optimally reduces the information loss and improves the test time performance eventually.

**DP-SGD and Linear Classifier with Handcrafted Features Beat DP-SGD on Deep Network:**
Tramer & Boneh (2021) showed that Differential Privacy has not reached its ImageNet moment yet, i.e. training end-to-end deep learning models do not outperform learning from Handcrafted features yet. To demonstrate this, they Papernot et al. (2021) experimentally show that a linear classifier on handcrafted features of the data extracted using ScatterNet Oyallon & Mallat (2015) outperforms then state-of-the-art DP-SGD approach. Scatter Network extracts features from images using wavelet transforms. We use the default parameters, depth $J = 2$ with rotations $L = 8$. For an image of size $(Ch, H, W)$, the Scatter network will result in features of shape $(81 * Ch, H/4, W/4)$. These features are flattened to a shape of $(81 * Ch * H * W/16, 1)$ to train a linear classifier and remain as is to train a CNN.

# 4 Analysis of `DP-SGD` in Federated Learning

In this section, we use DP-SGD in a federated learning setting to ensure data privacy at the example level. We first define the setting and the threat model. Adapting analysis from `DP-SGD` along with applying the parallel composition theorem, we can claim the desired $(\epsilon, \delta)$ DP guarantees. Designing noise to achieve $(\epsilon, \delta)$-DP is not a challenge with the simple observation (Observation 1). The main focus is to analyze the performance dependence on the frequency of aggregation, i.e., distribution of $T$ into local epochs per round of global epoch (where aggregation happens) to achieve the best privacy-accuracy trade-offs. Towards this, we prove the main result of the paper: for a given privacy budget and a fixed $T$, our theorem says, $E_r = 1, R = T$ offers the best performance. We finally show that the learned model's utility increases with the number of clients $k$ for a fixed privacy budget.

## 4.1 Private Federated Learning (PFL) Using DP-SGD

Federated learning setting under study has a set of $k$ clients $\{c_1, c_2, \ldots, c_k\}$ and server $S$. Learning happens in $R$ global rounds. In round $r$ of global training, (1) $S$ shares parameters $\boldsymbol{\theta}^{r-1}$ of a homogeneous model $M^{r-1}$ to each client $c_i$, (2) $c_i$ trains the model $M^r$ by initializing it using $M_i^{r-1}$. The training is done locally at $c_i$ for $E_r$ epochs using training set $D_i$ with `DP-SGD` Abadi et al. (2016) to obtain $M_i^r$ (with parameters $\boldsymbol{\theta}_i^r$). (3) Each client $c_i$ sends $\boldsymbol{\theta}_i^r$ to server $S$ and it uses `FedAvg` to obtain the aggregated model $M^r$ with parameters $\boldsymbol{\theta}^r = \frac{1}{k} \sum_{i=1}^{k} \boldsymbol{\theta}_i^r$. The setting is discussed in detail in Algorithm 1.

**Threat Model.** We assume an adversarial server $S$ as shown in Figure 1. In any global round $r$, client $c_i$ trains model $M_i^r$ and shares it with server $S$. Individual data $D_i$ for client $c_i$ is privately held with the client. However, trained model parameters $M_i^r$ and aggregated model $M^r$ are exposed to the server. Due to attacks such as Model-inversion Fredrikson et al. (2015), the trained models may leak information about individual clients' data to the server. Messages passing through the communication channels are publicly visible, i.e., can be observed by an external adversary.

## 4.2 Privacy Analysis

In this section, we give privacy guarantees of the PFL algorithm described in Algorithm 1. We make the following assumptions in our analysis.

**Assumption 1 ($\beta-$Lipschitz Property of Loss Function)** *We assume that the loss function $\mathcal{L}$ is $\beta$-lipschitz w.r.t. the client network parameters $\boldsymbol{\theta}$ (Virmaux & Scaman (2018)). That is, $\exists\ c \in \mathbb{R}^+$ such that*

---

**Algorithm 1** Private Federated Learning (PFL) Algorithm

---

**Input:** # Global rounds $R$; $E_1, \ldots, E_R$ where $E_r$ is # local epochs corresponding to global round $r$; datasets $D_1, \ldots, D_k$; noise variance $\sigma^2$; gradient clipping parameter $C$; mini-batch size $L$.

**Initialize:** $[\boldsymbol{\theta}_1^{1,1}, \boldsymbol{\theta}_2^{1,1}, \ldots \boldsymbol{\theta}_k^{1,1}]$

**for** $r \in 1 \ldots R$ **do**             ▷ Global round $r$

    **for** clients $i = 1 \ldots k$ **do**

        **for** $t \in 1 \ldots E_r^\dagger$ **do**          ▷ Local training for client $i$

            Randomly sample a mini-batch $B_i^t$ of size $L$ from $D_i$

            Intialize Grad-Sum $= \mathbf{0}$

            **for** $(\mathbf{x}, y) \in B_i^t$ **do**

                $\mathbf{g}(\mathbf{x}, y) \leftarrow \nabla_{\boldsymbol{\theta}_i^r} \mathcal{L}(\boldsymbol{\theta}_i^r; (\mathbf{x}, y))$         ▷ Per sample gradients

                $\mathbf{g}(\mathbf{x}, y) \leftarrow \mathbf{g}(\mathbf{x}, y) / \max(1, \frac{\|\mathbf{g}(\mathbf{x},y)\|_2}{C})$     ▷ Gradient Clipping

                Grad-Sum $=$ Grad-Sum $+ \mathbf{g}(\mathbf{x}, y)$

            **end for**

            $\tilde{\boldsymbol{\theta}}_i^{r,t+1} \leftarrow \tilde{\boldsymbol{\theta}}_i^{r,t} - \frac{\alpha}{L}\left(\text{Grad-Sum} + \mathcal{N}(0, \sigma^2 C^2 \boldsymbol{I})\right)$    ▷ Parameter update using DP-SGD

        **end for**

    **end for**

    $\boldsymbol{\theta}^r = \frac{1}{k}\sum_{i=1}^k \boldsymbol{\theta}_i^{r,E_r+1}$            ▷ FedAvg

    $\boldsymbol{\theta}_i^{r+1,1} = \boldsymbol{\theta}^r \;\; \forall i \in [k]$

**end for**

Return $\boldsymbol{\theta}^R$

---

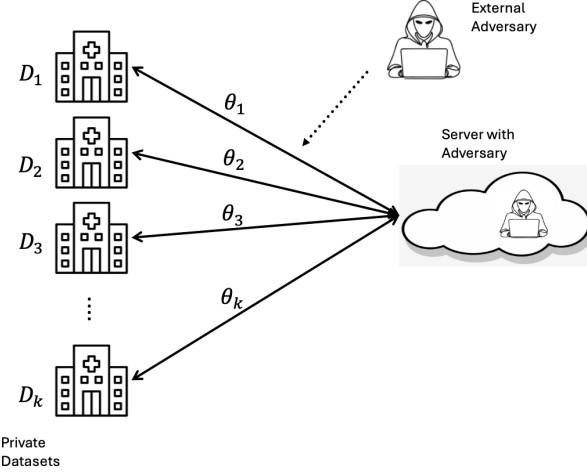

Figure 1: Threat Model

$\forall \; \boldsymbol{\theta}, \boldsymbol{\theta}' \in \Theta$, *we have:*

$$|\mathcal{L}(\boldsymbol{\theta}; (\mathbf{x}, y)) - \mathcal{L}(\boldsymbol{\theta}'; (\mathbf{x}, y))| \leq c\|\boldsymbol{\theta} - \boldsymbol{\theta}'\|, \quad \forall (\mathbf{x}, y) \in \mathcal{X} \times \mathcal{Y}$$

.

**Main Claim:** The DP guarantees that the model that is trained by a client $c_i$ for $T$ epochs (which happens in $R$ global rounds, each containing $E$ local epochs of training) is at least as good as a model trained locally for $T$ epochs on the same dataset. We establish this claim from the following simple observation.

**Observation 1** *The privacy budget $(\epsilon_{int}, \delta_{int})$ of any intermediate epoch in a $(\epsilon, \delta)$-DP-SGD trained algorithm is $(\epsilon_{int}, \delta_{int})$ then, $\epsilon_{int} \leq \epsilon$ and $\delta_{int} \leq \delta$.*

For DP-SGD to achieve $(\epsilon, \delta)$-differential privacy, Abadi et al. (2016) proved that the algorithm should add Gaussain noise with minimum standard deviation $\sigma_{min} = \kappa \frac{q\sqrt{T \log(1/\delta)}}{\epsilon}$ in each component. Here, $q$ is the sampling probability, $T$ denotes the number of training steps, and $\kappa$ is a constant factor. On training our model for a smaller number of rounds, $T_{\text{int}} < T$, where $\gamma = T/T_{\text{int}}$. The noise added will be sampled from the distribution $\mathcal{N}(\mathbf{0}, \sigma_{min}^2 \boldsymbol{I})$.

This implies that the privacy budget $(\epsilon_{\text{int}}, \delta)$ of any intermediate epoch in a $(\epsilon, \delta)$-DP-SGD trained algorithm is necessarily stronger than the final result, i.e., $\epsilon_{\text{int}} = \frac{\epsilon}{\sqrt{\gamma}} \leq \epsilon$. This holds, since $\gamma$ is greater than 1.

We analyze privacy in an FL-based training where each client trains their local model using DP-SGD with guarantees of $(\epsilon, \delta)$ DP in $T$ epochs. We conclude that in such a case, any intermediate model communicated to the server satisfied $(\epsilon, \delta)$-DP for each dataset $D_i$ corresponding to the client $c_i$ We formalize this result through Theorem 1.

**Claim 1** *PFL satisfies $(\epsilon, \delta)$-differential privacy.*

**Proof 1** *According to (Abadi et al., 2016, Theorem 1) and Lemma 1, we establish that the client model (before aggregation) guarantees a level of privacy at least as stringent as that promised by the* `DP-SGD` *mechanism. Furthermore, given that each intermediate model during training exhibits higher privacy than the final model, the privacy criteria is consistently met at each global aggregation round for all $c_i$. Additionally, from (Smith et al., 2021, Theorem 1), since the datasets for each client are modelled as I.I.D. samples from $\mathcal{D}$, we know that the global aggregated model is also at least as private as any of its ingredient client models.*

This result shows privacy budget remains intact for each client dataset post `FedAvg` step. We next discuss how a more frequent `FedAvg` step increases the model performance without compromising privacy.

### 4.3 Effect of number of local epochs on Performance

**Motivation — Budget on the Number of Gradient Updates.** For any Machine Learning technique, we would like to train for a sufficiently large number of epochs (gradient updates) to have a good model. However, doing so increases the exposure of the Model to data points. Due to this increased exposure to the data, more noise needs to be added to ensure $(\epsilon, \delta)$ privacy budget. DP-SGD adds noise to each gradient update step, and this noise is dependent on the total gradient updates $T$ (total epochs). From Abadi et al. (2016), we know that the privacy budget is directly proportional to $\sqrt{T}$. The standard deviation of noise added to each gradient update is $\sigma$ then from (Abadi et al., 2016, Equation 1) we have: $\sigma \cdot \epsilon \propto \sqrt{T}$.

Therefore, to increase $T$ — (1) Either privacy budget will be compromised or (2) Accuracy will be compromised because of higher noise added to each gradient update. Thus, `DP-SGD` has a budgeted number of gradient update epochs $T$. In FL, total gradient updates are divided into $R$ global rounds each $r \in [1, R]$ consists of $E_r$ local gradient updates; the budget $T = \sum_{r=1}^{R} E_r$ still preserved. In this section, we find the ideal distribution of $T$ across global rounds in privacy-preserving FL[3]. In other words, we provide an optimal number for local epochs in each global round. In the next theorem, we are giving a formal proof for optimal choice for $E_r$.

**Theorem 2** $E_r = 1$ *is performance optimal strategy in PFL.*

Due to space constraints, we defer the complete proof to Appendix A. We provide a rough proof sketch below. We consider two methods of training. **Method 1** is ensuring $(\epsilon, \delta)-$differential privacy, i.e. using `DP-SGD` Abadi et al. (2016) for local updates and `FedAvg` for global rounds of aggregation. **Method 2** trains the model using standard SGD for local updates and `FedAvg` for global rounds of aggregation. Our goal is to minimize the difference in performance degradation, which happens in `DP-SGD` due to noise addition. Towards this, we consider $E$ local epochs for $R$ global rounds such that $E \cdot R = T$ is fixed. On analysis, we find the minima is at $E = 1$ for $E \in \mathbb{N}^+$ (set of non-zero natural numbers).

---

[3]Although the provided analysis is for DP-SGD, the authors believe this result can be extended for other privacy-preserving learning techniques where two conditions are satisfied: (1) $\epsilon \cdot \sigma \propto Poly(T)$ and (2) noise is added during gradient update

To summarize, whenever training is budgeted in the number of local updates, it is always higher utility to have 1 local update per global round than any other strategy. We further back our results through experiments in Section 6.1. Having shown the (1) robustness of `DP-SGD` with `FedAvg`, and (2) optimal split of $T$ gradient updates over global rounds of training, we show in the following section the effect of several clients in the performance of privacy-preserving FL models.

### 4.4 Effect of the number of clients

We now analyse the effect of the number of clients on the performance of the model. Towards this, we first define the utility of the server's aggregated model below.

**Definition 2** ($l-$**Utility**) *Utility for the model with parameters $\boldsymbol{\theta}$, ideal set of parameters $\boldsymbol{\theta}^*$ and some dataset $D \subset \mathcal{X}$ sampled from distribution $\mathcal{D}$ is defined for some $l \in [0, 1]$ as*

$$U_l(\boldsymbol{\theta}, \boldsymbol{\theta}^*, \mathcal{D}) := \mathbb{P}(\mathbb{E}[|\mathcal{R}(\boldsymbol{\theta}, D) - \mathcal{R}(\boldsymbol{\theta}^*, D)|] < l)$$

We use this utility equation to show that as the number of clients increases, the utility of the model also increases. More specifically, the utility of the model changes by $O\left((1 - \frac{1}{k})\right)$ with $k$ clients. We formally prove this result in Theorem 3.

**Theorem 3** *For a $(\epsilon, \delta)-$differentially private FL model with local training using `DP-SGD` and aggregation using `FedAvg`, trained for $T$ total epochs and $k$ clients, we have $U_l(\boldsymbol{\theta}, \boldsymbol{\theta}^*, \mathcal{D}) \sim 1 - O\left(\frac{1}{l^2 k}\right)$.*

**Proof 2** *Consider any global round $r$ of training. In this global round, client $c_i$ initialize its model with $M^{r-1}$ independently on their data $D_i$ and outputs the model $M_i^r$ with parameter $\boldsymbol{\theta}_i^r$. These parameters $\boldsymbol{\theta}_i^r$ can be viewed as random variables sampled from some distribution $\mathbb{Q}(\boldsymbol{\theta}^*, \sigma)$. In the `FedAvg` step, we update the parameters of the global model to be $\boldsymbol{\theta}^r = \frac{1}{k} \sum_{i=1}^{k} \boldsymbol{\theta}_i^r$. These parameters are random variables following some distribution, with the mean as $\boldsymbol{\theta}^*$. Hence, $\boldsymbol{\theta}^r \sim \mathbb{Q}(\boldsymbol{\theta}^*, \frac{\sigma}{\sqrt{k}})$.*

*By Assumption 1 we have $|\mathcal{L}(\boldsymbol{\theta}, D) - \mathcal{L}(\boldsymbol{\theta}^*, D)| \leq \beta ||\boldsymbol{\theta} - \boldsymbol{\theta}^*||$. Thus,*

$$\mathbb{P}\left(\mathbb{E}[|\mathcal{R}(\boldsymbol{\theta}, D) - \mathcal{R}(\boldsymbol{\theta}^*, D)|] \geq l\right) \leq \mathbb{P}\left(\mathbb{E}[||\boldsymbol{\theta} - \boldsymbol{\theta}^*||] \geq \frac{l}{\beta}\right) \quad \leq \frac{\sigma^2 \beta^2}{l^2 k}$$

*We obtain the last equation through Chebyschev's inequality Mitzenmacher & Upfal (2017). Therefore, by definition of utility Definition 2, we have $U_l(\boldsymbol{\theta}, \boldsymbol{\theta}^*, \mathcal{D}) \sim 1 - O(\frac{1}{l^2 k})$.*

We therefore show that with a large number of clients, the server obtains a better performing model while guaranteeing the same privacy to each client dataset.

Having proved some essential results about role of DP in FL, now we study the effect of splitting $T$ training epochs as $E \times R$, $R$ being the number of global rounds and $E$ being the number of local training epochs per global round on performance of the model and also the effect of number of clients w.r.t. privacy budget and performance in the next section.

## 5 Evaluation of PFL on Real-World Dataset

First, we start with an experimental setup.

### 5.1 Experimental Set-up

**Datasets Used:** We perform experiments on three common benchmarks for Differentially Private ML: MNIST LeCun & Cortes (2010), Fashion-MNIST Xiao et al. (2017), CIFAR-10 Krizhevsky et al. (2009).

**Types of Network Architectures Used at Clients in PFL (Algorithm 1):** We analyze our findings by using four different kinds of network architectures for clients.

1. **CNN+ReLU:** In this model, we use CNN architecture presented in Papernot et al. (2021) with ReLU activation.

2. **CNN+TS:** In this model, we use CNN architecture presented in Papernot et al. (2021) with tempered sigmoid (TS) activation with $s = 2, temp = 2, o = 1$ (tanh).

3. **SN+Linear:** In the third model, we used features extracted from ScatterNet (SN) Oyallon & Mallat (2015) with its default parameters of depth=2 and rotations=8. We extract Scatternet features of shape (81,7,7) for MNIST and FashionMNIST datasets. For CIFAR10, we extracted features of shape (243,8,8). We flattened these features and trained a linear classifier.

4. **SN+CNN:** Here, we use the Scatternet features described in model M3. But, instead of training a linear model, we use CNN architecture described in Tramer & Boneh (2021) using group normalization.

**Experiments**

**Effect of $E$ on accuracy** We train one or more of the above NN architectures on the listed datasets for different values of $(E, R)$ combinations.

**Effect of $k$ on accuracy** We train one or more of the above NN architectures on the listed datasets for different values of $(T, epsilon)$ combinations.

### 5.2 Training

We trained Differentially Private ML Models uding PyTorch Opacus PyTorch (2023). We use `DP-SGD` , by adding sampled noise to gradients of each mini-batch to achieve Local Differential Privacy. To calculate the amount of noise required to add to each mini-batch, we use the `DP-SGD` formulation with a slight variation. We consider the number of total epochs as the product of local epochs in each client and the number of global rounds of training. We split the dataset by sampling 10% (6k for MNIST and FashionMNIST, 5k for CIFAR10) of random samples from the training set for each client. All the clients participate in each global epoch. We used $C = 1.0$ for all 3 datasets. We trained using SGD Optimizer with a learning rate of 0.3 and momentum of 0.5. We presented results $\varepsilon = 2.93, 2.7, 7.52$ for MNIST, FashionMNIST and CIFAR10 from previous work Papernot et al. (2021) and also a comparatively tighter bound to observe results following our claims.

**Evaluation** We trained federated models and compared the accuracy of the model after the final global round. We trained each setting for 10 runs and reported the average accuracy. For analyzing 2, we trained the model of $k = 10$ and set $T = 20$ and varied all possible settings of $(E, R)$. For analyzing 3 we set $E = 1, R = 20$ and compared performance of models for $k = \{10, 25, 50, 100\}$.

## 6 Observations

In this section, we demonstrate the validity of our analysis through empirical evaluation of real-world datasets. We first explain the experimental setup, followed by our results and discussion.

### 6.1 Effect of local epochs

In the federated setting, we explored different combinations of local and global epochs and evaluated which combination results in the best model performance for a fixed privacy budget to validate Theorem 2. A fixed $T$ will mean noise is sampled from the distribution of equal variance to each mini-batch irrespective of the

| MNIST | | | | | | | | | | | | |
|---|---|---|---|---|---|---|---|---|---|---|---|---|
| | FedAvg | | | | PFL With $\epsilon = 2.93$ | | | | PFL With $\epsilon = 1.2$ | | | |
| $(E, R)$ | CNN+RL | CNN+TS | SN+Lin | SN+CNN | CNN+RL | CNN+TS | SN+Lin | SN+CNN | CNN+RL | CNN+TS | SN+Lin | SN+CNN |
| $T = 20$ | | | | | | | | | | | | |
| $(1, 20)$ | 98.20 | 97.94 | 98.99 | 98.58 | 93.86 | 93.56 | 97.52 | 95.34 | 91.30 | 90.56 | 96.42 | 93.83 |
| $(2, 10)$ | 97.99 | 97.92 | 99.01 | 98.61 | 92.63 | 93.03 | 97.52 | 95.17 | 89.89 | 90.64 | 96.86 | 90.97 |
| $(4, 5)$ | 96.89 | 98.04 | 99.04 | 98.23 | 91.39 | 91.79 | 97.59 | 94.00 | 77.89 | 88.09 | 96.71 | 87.65 |
| $(5, 4)$ | 97.75 | 97.90 | 98.84 | 98.17 | 91.39 | 90.75 | 97.34 | 91.31 | 82.09 | 86.41 | 96.60 | 82.41 |
| $(10, 2)$ | 96.55 | 97.36 | 98.84 | 97.84 | 79.53 | 85.38 | 97.42 | 87.65 | 48.19 | 76.88 | 95.79 | 64.38 |
| $(20, 1)$ | 9.74 | 95.37 | 98.64 | 92.24 | 41.60 | 49.75 | 96.85 | 36.22 | 13.24 | 21.65 | 94.57 | 16.79 |

Table 2: Accuracy of PFL model on MNIST for a fixed $T$ with different combinations of $(E, R)$.

| Fashion-MNIST | | | | | | | | | | | | |
|---|---|---|---|---|---|---|---|---|---|---|---|---|
| | FedAvg | | | | PFL With $\epsilon = 2.7$ | | | | PFL With $\epsilon = 1.2$ | | | |
| $(E, R)$ | CNN+RL | CNN+TS | SN+Lin | SN+CNN | CNN+RL | CNN+TS | SN+Lin | SN+CNN | CNN+RL | CNN+TS | SN+Lin | SN+CNN |
| $T = 20$ | | | | | | | | | | | | |
| $(1, 20)$ | 86.11 | 86.54 | 90.01 | 88.28 | 78.41 | 80.14 | 86.01 | 81.26 | 75.78 | 77.34 | 84.96 | 79.03 |
| $(2, 10)$ | 85.88 | 86.69 | 90.07 | 88.35 | 77.29 | 78.98 | 86.46 | 80.16 | 74.56 | 76.34 | 84.77 | 79.08 |
| $(4, 5)$ | 85.15 | 86.37 | 90.13 | 87.47 | 71.74 | 77.07 | 86.13 | 78.40 | 67.56 | 74.12 | 84.78 | 74.92 |
| $(5, 4)$ | 85.73 | 86.02 | 90.28 | 87.83 | 71.54 | 76.44 | 86.33 | 77.43 | 69.33 | 73.87 | 84.74 | 70.63 |
| $(10, 2)$ | 84.84 | 84.86 | 90.38 | 86.23 | 62.49 | 74.34 | 85.22 | 68.95 | 34.34 | 64.43 | 84.59 | 55.37 |
| $(20, 1)$ | 29.58 | 81.34 | 89.61 | 81.58 | 39.52 | 55.36 | 83.73 | 44.12 | 9.17 | 58.68 | 82.79 | 38.19 |

Table 3: Accuracy of PFL model on Fashion-MNIST for a fixed $T$ with different combinations of $(E, R)$.

combination of local epochs and global rounds. We presented the results for different datasets in Table 2, Table 3 and Table 4. We observe the following patterns which are common results for all three datasets.

- **Frequent averaging $\implies$ Better performance:** We can observe (from that for a fixed $T$, out of every strategy used, aggregating local models every local epoch i.e. $(E, R) = (1, T)$ always results in the best performance as claimed in Theorem 2 across all methods. While this phenomenon does not hold in noise-free model aggregation, we can observe in Table 2 for a fixed privacy budget the aggregated model accuracy drops from $91.30\% \to 13.24\%$ by changing $(1, T)$ to $(T, 1)$ epoch split in MNIST for Vanilla `DP-SGD`.

- **Performance-Communication tradeoff:** There is a trade-off in increased communication complexity when $E = 1$. This complexity can be reduced by choosing a higher $E$ at the cost of

| | CIFAR-10 | | | | | | | | | | | |
| --- | --- | --- | --- | --- | --- | --- | --- | --- | --- | --- | --- | --- |
| | FedAvg | | | | PFL With $\epsilon = 7.53$ | | | | PFL With $\epsilon = 3.0$ | | | |
| $(E, R)$ | CNN+RL | CNN+TS | SN+Lin | SN+CNN | CNN+RL | CNN+TS | SN+Lin | SN+CNN | CNN+RL | CNN+TS | SN+Lin | SN+CNN |
| $T = 20$ | | | | | | | | | | | | |
| $(1, 20)$ | 48.15 | 47.49 | 61.05 | 63.84 | 38.98 | 40.25 | 54.58 | 52.40 | 29.72 | 34.46 | 51.08 | 45.82 |
| $(2, 10)$ | 42.57 | 44.78 | 61.62 | 64.04 | 33.64 | 36.51 | 55.02 | 48.35 | 17.54 | 29.22 | 51.28 | 41.50 |
| $(4, 5)$ | 37.52 | 38.29 | 62.82 | 61.98 | 18.07 | 27.46 | 55.07 | 41.54 | 14.25 | 20.52 | 50.94 | 33.93 |
| $(5, 4)$ | 34.25 | 36.06 | 63.70 | 59.95 | 16.04 | 20.93 | 53.90 | 42.48 | 12.14 | 20.83 | 51.35 | 31.75 |
| $(10, 2)$ | 19.24 | 18.14 | 62.63 | 51.21 | 12.15 | 18.70 | 55.27 | 30.66 | 10.00 | 14.21 | 51.82 | 16.95 |
| $(20, 1)$ | 16.05 | 15.48 | 61.91 | 25.33 | 10.00 | 17.81 | 54.00 | 19.65 | 10.00 | 12.74 | 49.13 | 10.78 |

Table 4: Accuracy of PFL model on CIFAR-10 for a fixed $T$ with different combinations of $(E, R)$.

| | MNIST ($\epsilon = 2.93$) | | | | Fashion-MNIST ($\epsilon = 2.7$) | | | | CIFAR-10 ($\epsilon = 7.53$) | | | |
| --- | --- | --- | --- | --- | --- | --- | --- | --- | --- | --- | --- | --- |
| $k$ | CNN+RL | CNN+TS | SN+Lin | SN+CNN | CNN+RL | CNN+TS | SN+Lin | SN+CNN | CNN+RL | CNN+TS | SN+Lin | SN+CNN |
| 10 | 93.06 | 93.56 | 97.52 | 95.34 | 78.41 | 80.14 | 86.01 | 81.26 | 38.98 | 40.25 | 54.58 | 52.4 |
| 25 | 93.59 | 94.61 | 98.14 | 96.00 | 78.74 | 80.18 | 87.74 | 82.78 | 41.63 | 40.69 | 58.46 | 53.99 |
| 50 | 94.83 | 94.63 | 98.22 | 96.82 | 78.86 | 80.63 | 88.13 | 83.49 | 40.10 | 41.72 | 60.58 | 54.85 |
| 100 | 95.08 | 95.03 | 98.41 | 96.46 | 78.89 | 80.69 | 88.20 | 83.78 | 40.91 | 43.36 | 62.66 | 55.15 |
| 1 | 95.28 | 95.63 | 98.42 | 96.81 | 80.97 | 84.11 | 88.34 | 85.18 | 55.86 | 57.31 | 63.01 | 61.86 |

Table 5: Accuracy of PFL model with varying number of clients for all datasets.

performance. However, in many practical applications (such as banks and hospitals) performance is usually preferred even at the cost of communication complexity.

- **Small Neural Networks are Robust:** ScatterNet + Linear method outperforms all methods across all datasets and $(E, R)$ combinations which abide by the results in Tramer & Boneh (2021). Along with that, the difference in performance between $(E, R) = (1, T)$ and $(E, R) = (T, 1)$ is very minimal for this method. While training end-to-end models on Fashion-MNIST as shown in Table 3 the accuracy is varied $78.4\% \rightarrow 39.52\%$ and $80.14\% \rightarrow 55.36\%$, this method performance varies from $86.01\% \rightarrow 82.79\%$ proving to be robust to epoch splits. This is due to the minimal number of parameters in these models. Even training a CNN on these same features drops the accuracy from $84.26\% \rightarrow 44.12\%$

## 6.2 Effect of number of clients

After we defined the number of clients and split the dataset among these clients, each client starts training local differentially private models. After each local model is trained, all the local models are aggregated using `FedAvg` and are tested on the global test dataset to check the model performance. We verify Theorem The-

orem 3 by training on a varying number of clients while keeping the rest of the parameters constant. We can observe an increase in the model performance of the aggregated model with an increase in the number of clients part of training for all values of $\epsilon$ Table 5. As $k$ increases, the model performance reaches closer to the model trained in the central setting.

## 7 Conclusion

We establish an optimal training protocol for training Local Differentially Private Federated Models. We show that communicating the local model to the server after every local epoch and increasing the number of clients participating in the process will improve the aggregated model performance. We back our findings with theoretical guarantees. We also validate our findings with experimental analysis of real-world datasets.

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

## A Proof of Theorem 2

Consider two methods of training at a client.

1. **PFL (Algorithm 1) DP-SGD + FedAvg:** This approach ensures $(\epsilon, \delta)-$differential privacy, i.e. using `DP-SGD` for local updates and `FedAvg` for global rounds of aggregation. Note that `DP-SGD` uses gradient clipping with clipping parameter $C$. For client $i$, corresponding model parameters are represented as $\tilde{\boldsymbol{\theta}}_i$.

2. **SGD + FedAvg:** It trains the model using SGD for local updates and `FedAvg` for global rounds of aggregation. SGD does not use gradient clipping. We treat no-gradient clipping as gradient clipping with a very high clipping parameter $C_1$. When clipping parameter is very high, we will never be clipping the gradient in SGD. For client $i$; corresponding parameters are $\boldsymbol{\theta}_i$.

Our goal is to find $E$ (the number of local epochs in each global epoch) in PFL Algorithm 1 such that the performance degradation due to noise addition is minimized. For client $i$, the performance degradation is captured as follows.

$$\Delta = \mathbb{E}[|\mathcal{R}(\tilde{\boldsymbol{\theta}}^{E+1}, D_i) - \mathcal{R}(\boldsymbol{\theta}^{E+1}, D_i)|]$$

where $\tilde{\boldsymbol{\theta}}_i^{E+1}$ be the parameters returned by DP-SGD for client $i$ after $E$ local epochs, $\boldsymbol{\theta}_i^{E+1}$ be the parameters returned by SGD for client $i$ after $E$ local epochs, $D_i$ be the training data corresponding to client $i$ and $\mathcal{R}(\boldsymbol{\theta}, D_i) = \frac{1}{|D_i|} \sum_{(\mathbf{x},y) \in D_i} \mathcal{L}(\boldsymbol{\theta}^{E+1}; (\mathbf{x}, y))$. The expectation is with respect to the product distribution $\mathcal{D}^{|D_i|} \times \prod_{t=1}^{E} \mathcal{N}(\mathbf{0}, \sigma^2 C^2 \mathbf{I})$. $\Delta$ can be further simplified as follows.

$$
\begin{aligned}
\Delta &= \frac{1}{|D_i|} \mathbb{E} \left[ \left\| \sum_{(\mathbf{x},y) \in D_i} \left( \mathcal{L}(\tilde{\boldsymbol{\theta}}^{E+1}, (\mathbf{x}, y)) - \mathcal{L}(\boldsymbol{\theta}^{E+1}, (\mathbf{x}, y)) \right) \right\| \right] \\
&\leq \frac{1}{|D_i|} \mathbb{E} \left[ \sum_{(\mathbf{x},y) \in D_i} \left| \mathcal{L}(\tilde{\boldsymbol{\theta}}^{E+1}, (\mathbf{x}, y)) - \mathcal{L}(\boldsymbol{\theta}^{E+1}, (\mathbf{x}, y)) \right| \right] \\
&\leq \frac{\beta}{|D_i|} \mathbb{E} \left[ \sum_{(\mathbf{x},y) \in D_i} \| \tilde{\boldsymbol{\theta}}^{E+1} - \boldsymbol{\theta}^{E+1} \| \right] \qquad \triangleright \text{ using Lipschitz property of } \mathcal{L} \\
&\leq \beta \mathbb{E} \left[ \| \tilde{\boldsymbol{\theta}}^{E+1} - \boldsymbol{\theta}^{E+1} \| \right]
\end{aligned}
$$

The gradient update step of SGD (local epoch in **SGD +FedAvg**) is

$$
\begin{aligned}
\boldsymbol{\theta}_i^{t+1} &= \boldsymbol{\theta}_i^t - \frac{\alpha_1}{L} \sum_{(\mathbf{x},y) \in B_i^t} \nabla \mathcal{L}(\boldsymbol{\theta}_i^t; (\mathbf{x}, y)) \frac{1}{\max\left(1, \frac{\|\nabla \mathcal{L}(\boldsymbol{\theta}_i^t; (\mathbf{x},y))\|}{C_1}\right)} \\
\Rightarrow \boldsymbol{\theta}_i^t - \boldsymbol{\theta}_i^{t+1} &= \frac{\alpha_1}{L} \sum_{(\mathbf{x},y) \in B_i^t} \nabla \mathcal{L}(\boldsymbol{\theta}_i^t; (\mathbf{x}, y)) \frac{1}{\max\left(1, \frac{\|\nabla \mathcal{L}(\boldsymbol{\theta}_i^t; (\mathbf{x},y))\|}{C_1}\right)}.
\end{aligned}
$$

Where $\alpha_1$ is the step size, $C_1$ is a the clipping parameters. As discussed, taking very large value of $C_1$ has same effect as no-clipping. $B_i^t \subset D_i$ be the minibatch presented to SGD in the $t^{th}$ local epoch at client $i$. Adding for all $t$ from 1 to $E$ where $E$ is the number of local epochs, we get

$$\boldsymbol{\theta}_i^1 - \boldsymbol{\theta}_i^{E+1} = \frac{\alpha_1}{L} \sum_{t=1}^{E} \sum_{(\mathbf{x},y) \in B_i^t} \nabla \mathcal{L}(\boldsymbol{\theta}_i^t; (\mathbf{x}, y)) \frac{1}{\max\left(1, \frac{\|\nabla \mathcal{L}(\boldsymbol{\theta}_i^t; (\mathbf{x},y))\|}{C_1}\right)}. \tag{1}$$

Similarly, for `DP-SGD` (local epoch in **DP-SGD + FedAvg**), using step-size $\alpha$, mini-batch size $L$, we have the following update equation.

$$\tilde{\boldsymbol{\theta}}_i^{t+1} = \tilde{\boldsymbol{\theta}}_i^t - \frac{\alpha}{L} \cdot \left( \sum_{(\mathbf{x},y) \in B_i^t} \nabla \mathcal{L}(\tilde{\boldsymbol{\theta}}_i^t; (\mathbf{x}, y)) \frac{1}{\max\left(1, \frac{\|\nabla \mathcal{L}(\tilde{\boldsymbol{\theta}}_i^t; (\mathbf{x},y))\|}{C}\right)} + \boldsymbol{\eta}_t \right).$$

Here, $\boldsymbol{\eta}_t \sim \mathcal{N}(\mathbf{0}, \sigma^2 C^2 \mathbf{I})$ and $\sigma$ depends on the privacy budget $\epsilon$ and parameter $\delta$ (see the details in (Abadi et al., 2016, Theorem 1)). Here, note that we take the same mini-batches in $E$ epochs that were used in **SGD + FedAvg**. Summing up all the terms for $t$ from 1 to $E$ we get,

$$\tilde{\boldsymbol{\theta}}_i^1 - \tilde{\boldsymbol{\theta}}_i^{E+1} = \frac{\alpha}{L} \sum_{t=1}^{E} \sum_{(\mathbf{x},y) \in B_i^t} \nabla \mathcal{L}(\tilde{\boldsymbol{\theta}}_i^t; (\mathbf{x}, y)) \frac{1}{\max\left(1, \frac{\|\nabla \mathcal{L}(\tilde{\boldsymbol{\theta}}_i^t; (\mathbf{x},y))\|}{C}\right)} + \frac{\alpha}{L} \sum_{t=1}^{E} \boldsymbol{\eta}_t. \tag{2}$$

Using the same initial parameters for SGD and DP-SGD, i.e., $\boldsymbol{\theta}_i^1 = \tilde{\boldsymbol{\theta}}_i^1$, we find $\|\tilde{\boldsymbol{\theta}}_i^{E+1} - \boldsymbol{\theta}_i^{E+1}\|$ as follows.

$$\|\tilde{\boldsymbol{\theta}}_i^{E+1} - \boldsymbol{\theta}_i^{E+1}\| \leq \frac{1}{L} \sum_{t=1}^{E} \sum_{(\mathbf{x},y) \in B_i^t} \left\| \frac{\alpha \nabla \mathcal{L}(\tilde{\boldsymbol{\theta}}_i^t; (\mathbf{x}, y))}{\max\left(1, \frac{\|\nabla \mathcal{L}(\tilde{\boldsymbol{\theta}}_i^t;(\mathbf{x},y))\|}{C}\right)} - \frac{\alpha_1 \nabla \mathcal{L}(\boldsymbol{\theta}_i^t; (\mathbf{x}, y))}{\max\left(1, \frac{\|\nabla \mathcal{L}(\boldsymbol{\theta}_i^t;(\mathbf{x},y))\|}{C_1}\right)} \right\| + \frac{\alpha}{L} \sum_{t=1}^{E} \|\boldsymbol{\eta}_t\|$$

$$\leq \frac{1}{L} \sum_{t=1}^{E} \sum_{(\mathbf{x},y) \in B_i^t} (\alpha C + \alpha_1 C_1) + \frac{\alpha}{L} \sum_{t=1}^{E} \|\boldsymbol{\eta}_t\| \qquad \triangleright \text{using triangle inequality}$$

In the above, we used the fact that $\left\| \frac{\alpha \nabla \mathcal{L}(\tilde{\boldsymbol{\theta}}_i^t;(\mathbf{x},y))}{\max\left(1, \frac{\|\nabla \mathcal{L}(\tilde{\boldsymbol{\theta}}_i^t;(\mathbf{x},y))\|}{C}\right)} \right\| \leq \alpha C$ and $\left\| \frac{\alpha_1 \nabla \mathcal{L}(\boldsymbol{\theta}_i^t;(\mathbf{x},y))}{\max\left(1, \frac{\|\nabla \mathcal{L}(\tilde{\boldsymbol{\theta}}_i^t;(\mathbf{x},y))\|}{C_1}\right)} \right\| \leq \alpha_1 C_1$. Taking

expectation on both sides with respect to the product distribution $\mathcal{D}^{|D_i|} \times \prod_{t=1}^{E} \mathcal{N}(\mathbf{0}, \sigma^2 C^2 \mathbf{I})$, we get the following.

$$\mathbb{E}\left[\|\tilde{\boldsymbol{\theta}}_i^{E+1} - \boldsymbol{\theta}_i^{E+1}\|\right] \leq \frac{1}{L} \sum_{t=1}^{E} \sum_{(\mathbf{x},y) \in B_i^t} (\alpha C + \alpha_1 C_1) + \frac{\alpha}{L} \sum_{t=1}^{E} \mathbb{E}[\|\boldsymbol{\eta}_t\|]$$

$$= E(\alpha C + \alpha_1 C_1) + \frac{\sqrt{2}\alpha\sigma C}{L} \frac{\Gamma(\frac{N+1}{2})}{\Gamma(\frac{N}{2})} = E\left(\alpha C + \alpha_1 C_1 + \frac{\sqrt{2}\alpha\sigma C}{L} \frac{\Gamma(\frac{N+1}{2})}{\Gamma(\frac{N}{2})}\right).$$

We use the fact that $\mathbb{E}[\|\boldsymbol{\eta}_t\|] = \sqrt{2}\sigma C \frac{\Gamma(\frac{N+1}{2})}{\Gamma(\frac{N}{2})}$, where $N$ is the size of the parameter vector $\boldsymbol{\theta}$. Thus, the performance degradation $\Delta$ is further bounded as follows.

$$\Delta \leq \beta \mathbb{E}[\|\tilde{\boldsymbol{\theta}}^{E+1} - \boldsymbol{\theta}^{E+1}\|] \leq \beta E\left(\alpha C + \alpha_1 C_1 + \frac{\sqrt{2}\alpha\sigma C}{L} \frac{\Gamma(\frac{N+1}{2})}{\Gamma(\frac{N}{2})}\right)$$

We see that the upper bound on $\Delta$ is minimum when $E = 1$.

