# OpenReview forum: "Optimal Strategies for Federated Learning  Maintaining Client Privacy"
_TMLR — Withdrawn by Authors_

### Review · Reviewer_GowL · 2025-03-04

**Summary Of Contributions:**

This paper studies privacy-preserving federated learning and suggests (1) aggregating once every epoch and (2) using a large number of clients are optimal. The authors provide both theoretical and empirical supports for the claim.

**Audience:**

Yes

**Broader Impact Concerns:**

N/A.

**Claims And Evidence:**

No

**Requested Changes:**

1. Revise the first claim on aggregation frequency, theorem 2 and its proof.

2. Provide concrete examples that would benefit from the second claim on client participation.

**Strengths And Weaknesses:**

**Strength**

This submission is in a good form. The claims and their support are clearly presented and are easy to follow.

**Weakness**
1. The first claim, "performance is proportional to the frequency of aggregation step. That means, the clients should update their local model to the server every local epoch for optimal performance", may not be entirely correct. If the performance is proportional to the frequency of aggregation step, the optimal strategy might be dp-sgd and aggregating every local step. I also read the proof in Appendix A to gain more insights. The proof uses $E$ to denotes the number of local epochs and shows that $E=1$ is optimal. However, in Equation (1), page 14, the role of the $E$ symbol only denotes the number of optimization steps instead of the number of epochs.

2. The second claim, "As the number of participating clients increases, the aggregate global model converges to that of its
non-private counterpart", is a straightforward application of Chebyshev's inequality and, therefore, may not contribute much to many practical federated learning applications. Theorem 3 supports the second claim and proves that the sample mean of the added gradient noise converge to the mean (i.e., 0) such that the noise does not affect the convergence trajectory. In real-world systems, we may not have many clients (e.g., in cross-silo settings) and have limited control over the client participation.

---

### Review · Reviewer_2fQ3 · 2025-03-16

**Summary Of Contributions:**

The question asked by the paper is the following: in a DP-FL framework, at each round, should a client divide their overall update into several local updates, with each update having their own independent noise. The answer is obviously now, by standard results on local vs central DP accuracy.

**Audience:**

No

**Claims And Evidence:**

No

**Requested Changes:**

No requested change---the contributions of the paper need be fundamentally changed.

**Strengths And Weaknesses:**

At the moment, I unfortunately have to recommend the paper for rejection without revision. Let me review the weaknesses of the paper below, which I believe are not fixable without a resubmission of a significantly modified version of the paper:
- The writing of the paper is very unclear, informal, and inaccurate. The paper has a lot of statements of the form "This idea brings a clever
realization that Federated averaging is a noise addition step" which are not only wrong, but are completely disconnected from the technical contributions of the paper. It also took me several reads to understand what the main question was due to the poor writing.
- The main question that is asked is trivial. Unsurprisingly, the authors show that using a simple epoch is better. This is effectively re-proving that the local model is worse than the central model for privacy
- The contribution is too narrow and not novel enough, and in many places, it is unclear what is new. Incorporating DP in FL is not new---there is a plethora of papers doing so that are not cited. Algorithm 1 seems to be presented as a contribution---this is just the standard DP-FL setup. The only real new theoretical results are those of Sections 4.3 and 4.4. Even then, the result of section 4.4 is not surprising and yet presented as novel. The result states, effectively, that having more data points makes privacy easier, which is common wisdom for central DP with averaging/a sensitivity of 1/n; the specific bound given there is also trivial and likely already known.
- Another unfixable without completely changing the paper is that the technical contribution is flawed and wrong. Theorem 2 is stated oddly informally, with the optimality metric not being defined, which raised alarms or me. The metric is hidden in the Appendix. The problem is that the authors do not provide a proof that the metric is optimized for $E_r = 1$; they prove than some *upper bound* on that metric is optimized at $E_r = 1$. Unless the metric is tight/a good approximation, this invalidates the theoretical results.

Given the lack of novelty as well as technical issues with the paper, I recommend rejection without revision.

---

### Review · Reviewer_9VDd · 2025-04-08

**Summary Of Contributions:**

This paper studies the trade-offs between model performance, privacy, and communication complexity in federated learning (FL) frameworks that employ differential privacy through DP-SGD. The authors analytically demonstrate that, given a fixed privacy budget, optimal model performance is achieved when clients communicate their locally updated models after each training epoch (Theorem 2). Additionally, they prove that increasing the number of participating clients improves the utility of the global aggregated model, approaching performance similar to a non-private centralized model (Theorem 3). The paper empirically validates these theoretical findings across standard benchmark datasets such as MNIST, Fashion-MNIST, and CIFAR-10, exploring different neural network architectures and DP settings, thereby contributing practical insights toward designing efficient and privacy-preserving federated learning systems.

**Audience:**

Yes

**Claims And Evidence:**

Yes

**Requested Changes:**

Please see my comments above.

**Strengths And Weaknesses:**

## Pros:
   - Differential privacy (DP) in FL is timely and relevant.
   - Claims made regarding privacy versus utility trade-offs and optimal aggregation strategies (local vs global epochs) are important contributions if proven rigorously.
   - The authors attempt a formal treatment of their key claims through clear theoretical statements (Theorems 2 and 3) and proofs provided (appendix for theorem 2).
   - Experimental validation provided on standard datasets (MNIST, Fashion-MNIST, CIFAR-10), supporting the theoretical claims with practical results.

---

## Cons:

1. The primary theoretical result (one local epoch per global round is optimal) is somewhat intuitive and already known from federated learning literature. The novelty seems limited as similar results have appeared in other FL+DP literature.

2. Theorem 2 Proof Issue:
     - The proof provided in Appendix A relies heavily on simplifications and upper-bound approximations.
     - Particularly, the step of applying the triangle inequality is quite loose:
       \[
       \| \tilde{\theta}_{E+1} - \theta_{E+1}\| \leq \frac{1}{L}\sum_{t=1}^{E}\sum_{(x,y)}(\alpha C + \alpha_1 C_1) + \frac{\alpha}{L}\sum_{t=1}^{E}\|\eta^t\|
       \]
     - The proof assumes an overly simplistic scenario and the bound derived is not tight, weakening its significance.
     - Another critical issue: the authors did not justify why they assumed the same mini-batches are presented in both DP and non-DP versions in the analysis. This is a nontrivial assumption that directly impacts their derived results.

3. Theorem 3 Proof Issue:
     - The authors use Chebyshev’s inequality loosely to derive a utility bound. This proof does not account properly for correlation and distributional assumptions between client datasets. They assume implicitly the existence of a central distribution (homogeneous distribution of data), which is not realistic in FL.
     - More problematic is the vague definition of utility and its practical interpretation. The authors’ definition of utility:
       \[
       U_l(\theta,\theta^*,D)=P(E[|R(\theta,D)-R(\theta^*,D)|]<l)
       \]
       lacks rigorous justification on why this probabilistic definition should represent real-world FL performance.

4.  Experiments, though conducted on standard datasets, miss an essential dimension of FL: heterogeneity across client data distribution. This severely limits the applicability of results. Absence of experiments under realistic non-i.i.d. scenarios reduces the practical significance of findings.

5.  Several explanations in the main paper could be clearer, especially the motivation behind the utility definition and the practical implications of their theorems.

6. The following works also used DP-SGD in FL settings. The authors fail to mention these works: [1,2].


7. There quite a few typos in the paper among which I mention some here:
- There is a jump between the two paragraphs in the abstract. It is not very clear how the first sentence of the second paragraph is related to the first paragraph. I needed to read the introduction to understand what the authors aim to do. Please make the flow of abstract better.
- Abstract, line 6: "inferencing attacks" → "inference attacks".
- Page 1, Introduction, line 2: "over data distributed among" → "on data distributed among".
- Federated learning (FL) is abbreviated at least twice in the paper.
- DP is not defined in the intro (it is not abbreviated)
- DP-SGD is not defined in the intro. what does it stand for?
- Page 3, footnote 2: "does not matter if it is random or based on some prior knowledge." (colloquial phrasing, better rephrased as "whether initialized randomly or based on prior knowledge").
- Page 6, Proof of Claim 1: "consistently met at each global aggregation round for all ci." → "for all clients ci."
- Page 7, Motivation paragraph: "Thus, DP-SGD has a budgeted number of gradient update epochs T." → "DP-SGD imposes a limit (budget) on the number of gradient update epochs T."


**Given the identified issues, particularly limited technical novelty, loose theoretical analysis, unjustified assumptions in proofs (especially Theorems 2 and 3), limited experimental settings (no heterogeneous distributions), and some clarity issues,  I cannot recommend accepting the paper. The authors should revise the paper, and add more contributions to the current format.**


-----
1. Du, J., Li, S., Chen, X., Chen, S., & Hong, M. (2021). Dynamic differential-privacy preserving sgd. arXiv preprint arXiv:2111.00173.
2. Pustozerova, A., Baumbach, J., & Mayer, R. (2023, December). Differentially Private Federated Learning: Privacy and Utility Analysis of Output Perturbation and DP-SGD. In 2023 IEEE International Conference on Big Data (BigData) (pp. 5549-5558). IEEE.

---

### Note · Authors · 2025-04-22

I have read and agree with the venue's withdrawal policy on behalf of myself and my co-authors.